# Experimental Design for Learning Causal Graphs with Latent Variables

**Murat Kocaoglu**[*]
Department of Electrical and Computer Engineering
The University of Texas at Austin, USA
mkocaoglu@utexas.edu

**Karthikeyan Shanmugam**[*]
IBM Research NY, USA
karthikeyan.shanmugam2@ibm.com

**Elias Bareinboim**
Department of Computer Science and Statistics
Purdue University, USA
eb@purdue.edu

## Abstract

We consider the problem of learning causal structures with latent variables using interventions. Our objective is not only to learn the causal graph between the observed variables, but to locate unobserved variables that could confound the relationship between observables. Our approach is stage-wise: We first learn the observable graph, i.e., the induced graph between observable variables. Next we learn the existence and location of the latent variables given the observable graph. We propose an efficient randomized algorithm that can learn the observable graph using $\mathcal{O}(d \log^2 n)$ interventions where $d$ is the degree of the graph. We further propose an efficient deterministic variant which uses $\mathcal{O}(\log n + l)$ interventions, where $l$ is the longest directed path in the graph. Next, we propose an algorithm that uses only $\mathcal{O}(d^2 \log n)$ interventions that can learn the latents between both non-adjacent and adjacent variables. While a naive baseline approach would require $\mathcal{O}(n^2)$ interventions, our combined algorithm can learn the causal graph with latents using $\mathcal{O}(d \log^2 n + d^2 \log(n))$ interventions.

## 1 Introduction

Causality shapes how we view, understand, and react to the world around us. It is arguably a key ingredient in building intelligent systems that are autonomous and can act efficiently in complex environments. Not surprisingly, the task of automating the learning of cause-and-effect relationships have attracted great interest in the artificial intelligence and machine learning communities. This effort has led to a general theoretical and algorithmic understanding of the assumptions under which cause-and-effect relationships can be inferred from data. These results have started to percolate through the applied fields ranging from genetics to medicine, from psychology to economics [5, 26, 33, 25].

The endeavour of algorithmically learning causal relations may have started from the independent discovery of the IC [35] and PC algorithms [33], which almost identically, and contrary to previously held beliefs, showed the feasibility of recovering these relations from purely observational, non-experimental data. A plethora of methods followed this breakthrough, and now we understand, at least in principle, the limits of what can be inferred from purely observational data, including (not exhaustively) [31, 14, 21, 27, 19, 13]. There are a number of assumptions that have been considered about the data-generating model when attempting to unveil the causal structure. One of the most

---

[*]Equal contribution.

popular assumptions is that the data-generating model is *causally sufficient*, which means that no latent (unmeasured) variable affects more than one observed variable. In practice, this is a very stringent condition since the existence of latents affecting more than one observed variable, and generating what is called *confounding bias*, is one of the main concerns of empirical scientists. The problem of causation is deemed challenging in most of the empirical fields because scientists recognize that not all the variables influencing the observed phenomenon can be measured. The general question that arises is then how much of the observed behavior of the system is truly causal, or whether it is due to some external, unobserved forces [26, 5].

To account for the latent variables in the context of structural learning, the IC* [35] and FCI [33] algorithms were introduced, which showed the possibility of recovering causal structures *even when* latent variables may be confounding the observed behavior [2]. One of the main challenges faced by these algorithms is that although some ancestral relations as well as certain causal edges can be learned [36, 7], many observationally equivalent architectures cannot be distinguished. Despite the practical challenges when collecting the data (e.g., finite samples, selection bias, missing data), we now have a complete characterization of what structures are recoverable from observational data based on conditional independence constraints [33, 2, 37]. Inferences will be constrained within an equivalence class. Initial works leveraged ideas of experimental design and the availability of interventional data to move from the equivalence class to a specific graph, but almost exclusively considering causally sufficient systems [9, 15, 11, 12, 30, 18].

For causally insufficient systems, there is a growing interest in identifying experimental quantities and structures based on partially observed interventional data [4, 32, 29, 28, 24, 16, 8, 34, 22], but without the goal of designing the optimal set of interventions. Perhaps the most relevant paper to our setup is [23]. Authors identify the experiments needed to learn the causal graph under latents, given the output of FCI algorithm. However, they are not interested in minimizing the number of experiments.

In this paper, we propose the first efficient non-parametric algorithm for learning a causal graph with latent variables. It is known that $\log(n)$ interventions are necessary (across all graphs) and sufficient to learn a causal graph without latent variables [12], and we show, perhaps surprisingly, that there exists an algorithm that can learn any causal graph with latent variables which requires $\text{poly}(\log n)$ interventions when the observable graph is sparse. More specifically, our contributions are as follow:

- We introduce a deterministic [3] algorithm that can learn any causal graph and the existence and location of the latent variables using $\mathcal{O}(d \log(n) + l)$ interventions, where $d$ is the largest node degree and $l$ is the longest directed path of the causal graph.
- We design a randomized algorithm that can learn the observable graph and all the latent variables using $\mathcal{O}(d \log^2(n) + d^2 \log(n))$ interventions with high probability, where $d$ is the largest node degree.

The first algorithm is useful in practical settings where the longest directed path is not very deep, e.g., $\mathcal{O}(\log(n))$. This includes bipartite, time-series, and relational type of domains where the underlying causal topology is somewhat sparse. As an example application, consider the problem of inferring the causal effect of a set of genes on a set of phenotypes, that could be cast as learning a bipartite causal system. For the more general setting, we introduce a randomized algorithm that with high probability is capable of unveiling the true causal structure.

**Background**

We assume for simplicity that all the random variables are discrete. We use the language of Structural Causal Models (SCM) [26, pp. 204-207]. Formally, an SCM $\mathcal{M}$ is a 4-tuple $\langle \mathcal{U}, \mathcal{V}, \mathcal{F}, P(u) \rangle$, where $\mathcal{U}$ is a set of exogenous (unobserved, latent) variables, $\mathcal{V}$ is a set of endogenous (measured) variables. We partition the set of exogenous variables into two disjoint sets: Exogenous variables with one observable child, denoted by $\mathcal{E}$, exogenous variables with two observable children, denoted by $\mathcal{L}$. $\mathcal{F} = \{f_i\}$ is a collection of functions such that each endogenous variable $V_i \in \mathcal{V}$ is determined by a function $f_i \in F$: Each $f_i$ is a mapping from the respective domain of the exogenous variables associated with $V_i$ and a set of observable variables associated with $V_i$, called $PA_i$, into $V_i$. The

set of exogenous variables associated with $V_i$ can be divided into two classes, the one with a single observable child, denoted by $\mathcal{E}_i \in \mathcal{E}$, and those with two observable children, denoted by $\mathcal{L}_i \subseteq \mathcal{L}$. Hence $f_i$ maps from the domain of $\mathcal{E}_i \cup PA_i \cup \mathcal{L}_i$ to $V_i$. The entire set $\mathcal{F}$ forms a mapping from $\mathcal{U}$ to $\mathcal{V}$. The uncertainty is encoded through a product probability distribution over the exogenous variables $P(\mathcal{E}, \mathcal{L})$. For simplicity we refer to $\mathcal{L}$ as the set of latents, and $\mathcal{E}$ as the set of exogenous variables.

Within the structural semantics, performing an action $S = s$ is represented through the do-operator, $do(S = s)$, which encodes the operation of replacing the original equation of $S$ by the constant $s$ and induces a submodel $\mathcal{M}_S$ (also for when $S$ is not a singleton). We denote the post-interventional distribution by $P_S(\cdot)$. For a detailed discussion on the properties of structural models, we refer readers to [5, 23, 24, Ch. 7]. Define $D_\ell = (\mathcal{V} \cup \mathcal{L}, E_\ell)$ to be the causal graph with latents. We define the observable graph to be the induced subgraph on $\mathcal{V}$ which is $D = (\mathcal{V}, E)$.

In practice, we use an independent random variable $W_i$ taking values uniformly at random in the state space of $V_i$, to implement an intervention $do(V_i)$. A conditional independence statement, e.g., *X is independent from Y given $Z \subset \mathcal{V}$ with respect to causal model $\mathcal{M}_S$*, in shown by $(X \perp\!\!\!\perp Y | Z)_{\mathcal{M}_S}$, or $(X \perp\!\!\!\perp Y | Z)_S$ when the causal model is clear from the context. These conditional independencies are with respect to the post-interventional joint probability distribution $P_S(\cdot)$. In this paper, we assume that an oracle to conditional independence (CI) tests is available.

The *mutilated* or *post-interventional causal graph*, denoted $D_\ell[S] = (\mathcal{V} \cup \mathcal{L}, E_\ell[S])$, is identical to $D_\ell$ except that all the incoming edges incident on any vertex in the interventional set $S$ is absent, i.e., $E_\ell[S] = E_\ell - \{(Y, V) : V \in S, (Y, V) \in E_\ell\}$. We define the *transitive closure*, denoted $D_{\mathrm{tc}}$, of an observable causal DAG $D$ as follows: If there is a directed path from $V_i$ to $V_j$ in $D$, there is a directed edge from $V_i$ to $V_j$ in $D_{\mathrm{tc}}$. Essentially, a directed edge in $D_{\mathrm{tc}}$ represents an ancestral relation in $D$.

For any DAG $D = (V, E)$, a set of nodes $S \subset V$ d-separates two nodes $a$ and $b$ if and only if $S$ blocks all paths between $a$ and $b$. 'Blocking' is a graphical criterion associated with d-separation [4]. A probability distribution is said to be faithful (or stable) to a graph, if and only if every conditional independence statement can be read off from the graph using d-separation, see [26, Ch. 2] for a review. We assume that faithfulness holds in the observational and post-interventional distributions following [12].

**Results and outline of the paper**

The skeleton of the proposed learning algorithms can be split into 3 steps, namely:

$$\emptyset \xrightarrow{(a)} \text{Transitive Closure} \xrightarrow{(b)} \text{Observable graph} \xrightarrow{(c)} \text{Observable graph with Latent variables}$$

Each step requires different tools and graph theoretic concepts:

(a) We use a pairwise independence test under interventions that reveals the ancestral relations. This is combined in an efficient manner with separating systems to discover the transitive closure of $D$ in $\mathcal{O}(\log n)$ interventions.
(b) We rely on the transitive reduction of directed acyclic graphs that can be efficiently computed only from their transitive closure. A key property we observe is that the *transitive reduction reveals a subset of the true edges*. For our randomized algorithm, we use a sequence of transitive reductions computed from transitive closures (obtained using step (a)) of different post-interventional graphs.
(c) Given the observable graph, it is possible to discover latents between non-adjacent nodes using CI tests under suitable interventions. We use an edge-clique cover on the complement graph to optimize the number of experiments. For latents between adjacent nodes, we use a relatively unknown test called the do-see test, i.e., leveraging the equivalence between observing and intervening on the node. We implement it using induced matching cover of the observable graph.

The modularity of our approach allows us to solve subproblems: given the ancestral graph, we can use $(b)$ to discover the observable graph $D$. If $D$ is known, we can learn the latents with $(c)$. Some pictorial illustrations of the main results in the technical sections are found in the full version [20].

## 2 Identifying the Observable Graph: A simple baseline

We discuss a natural and a simple deterministic baseline algorithm that finds the observable graph with experiments when confounders are present. To our knowledge, a *provably complete* algorithm

that recovers the observable graph under this setting and is superior than this simple baseline in the worst case is not known. We start from the following observation. Suppose $X \to Y$ where $X, Y$ are observable variables and let $L$ be a latent variable such that $L \to X$, $L \to Y$. Consider the post interventional graph $D_\ell[\{X\}]$ where we intervene on $X$. It is easy to see that, $X$ and $Y$ are dependent in the post interventional graph too because of the direct causal relationship. However, if $X$ is not a parent of $Y$, then in the post interventional graph $D_\ell[\{X\}]$ even with or without the latent $L$ between $X$ and $Y$, $X$ is independent of $Y$ since $X$ is intervened on.

It is possible to recreate this condition between any target variable $Y$ and any one of its direct parents $X$ when many other observable variables are involved. Simply, we consider the post-interventional graph where we intervene on all observable variables but $Y$. In $D_\ell[V - \{Y\}]$, $Y$ and $X$ are dependent if and only if $X \to Y$ is a directed edge in the observable graph $D$, because every variable except $X$ becomes independent of all other variables in the post interventional graph. Therefore, one needs $n$ interventions, each of size $n-1$ to find out the parent set of every node. We basically show in the next two sections that when the graph $D$ has constant degree, it is enough to do $O(\log^2(n))$ interventions representing the first provably exponential improvement.

## 3 Learning Ancestral Relations

In this section, we show that *separating systems* can be used to construct sequences of pairwise CI tests to discover the transitive closure of the observable causal graph, i.e., the graph that captures all ancestral relations. The following lemma relates post-interventional statistical dependencies with the ancestral relations in the graph with latents.

**Lemma 1.** *[Pairwise Conditional Independence Test] Consider a causal graph with latents $D_\ell$. Consider an intervention on the set $S \subset \mathcal{V}$ of observable variables. Then, under the post-interventional faithfulness assumption, for any pair $X_i \in S, X_j \in \mathcal{V} \backslash S$, $(X_i \not\perp\!\!\!\perp X_j)_{D_\ell[S]}$ if and only if $X_i$ is an ancestor of $X_j$ in the post-interventional observable graph $D[S]$.*

Lemma 1 constitutes, for any ordered pair of variables $(X_i, X_j)$ in the observable graph $D$, a test for whether $X_i$ is an ancestor of $X_j$ or not. Note that a single test is not sufficient to discover the ancestral relation between a pair $(X_i, X_j)$, e.g., if $X_i \to X_k \to X_j$ and $X_i, X_k \in S, X_j \notin S$, the ancestral relation will not be discovered. This issue can be resolved by using a sequence of interventions guided by a separating system, and later finding the transitive closure of the learned graph.

Separating systems were first defined by [17], and has been subsequently used in the context of experimental design [10]. A separating system on a ground set $S$ is a collection of subsets of $S$, $\mathcal{S} = \{S_1, S_2 \ldots\}$ such that for every pair $(i, j)$, there is a set that contains only one, i.e., $\exists k$ such that $i \in S_k, j \notin S_k$ or $j \in S_k, i \notin S_k$. We require a stronger notion which is captured by a strongly separating system.

**Definition 1.** *An $(m, n)$ strongly separating system is a family of subsets $\{S_1, S_2 \ldots S_m\}$ of the ground set $[n]$ such that for any two pairs of nodes $i$ and $j$, there is a set $S$ in the family such that $i \in S$, $j \notin S$ and also another set $S'$ such that $i \notin S'$, $j \in S'$.*

Similar to separating systems, one can construct strongly separating systems using $\mathcal{O}(\log(n))$ subsets:

**Lemma 2.** *An $(m, n)$ strong separating system exists on a ground set $[n]$ where $m \leq 2\lceil \log n \rceil$.*

We propose Algorithm 1 to discover the ancestral relations between the observable variables. It uses the subsets of a strongly separating system on the ground set of all observable variables as intervention sets, to assure that the ancestral relation between every ordered pair of observable variables is tested. The following theorem shows the number of experiments and the soundness of Algorithm 1.

**Theorem 1.** *Algorithm 1 requires only $2\lceil \log n \rceil$ interventions and conditional independence tests on samples obtained from each post-interventional distribution and outputs the transitive closure $D_{\text{tc}}$.*

## 4 Learning the Observable Graph

We introduce a deterministic and a randomized algorithm for learning the observable causal graph $D$ from ancestral relations. $D$ encodes every direct causal connection between the observable nodes.

**Algorithm 1** LearnAncestralRelations- Given access to a conditional independence testing oracle (CI oracle), query access to samples from any post-interventional causal model derived out of $\mathcal{M}$ (with causal graph $D_\ell$), outputs all ancestral relationships between observable variables, i.e., $D_{\text{tc}}$

```
 1: function LEARNANCESTRALRELATIONS(M)
 2:     E = ∅.
 3:     Consider a strongly sep. system of size ≤ 2 log n on the ground set V - {S₁, S₂..S₂⌈log n⌉}.
 4:     for i in [1 : 2⌈log n⌉] do
 5:         Intervene on the set Sᵢ of nodes.
 6:         for X ∈ Sᵢ, Y ∉ Sᵢ, Y ∈ V do
 7:             Use samples from M_Sᵢ and use the CI-oracle to test the following.
 8:             if (X ⊥̸ Y)_Dℓ[S] then
 9:                 E ← E ∪ (X, Y).
10:             end if
11:         end for
12:     end for
13:     return The transitive closure of the graph (V, E)
14: end function
```

## 4.1 A Deterministic Algorithm

Based on Section 3, assume that we are given the transitive closure of the observable graph. We show in Lemma 3 that, when the intervention set contains all parents of $X_i$, the only variables dependent with $X_i$ in the post-interventional observable graph are the parents of $X_i$ in the observable graph.

**Lemma 3.** *For variable $X_i$, consider an intervention on $S$ where $Pa_i \subset S$. Then $\{X_j \in S : (X_i \not\perp X_j)_{D[S]}\} = Pa_i$.*

Let the longest directed path of $D_{\text{tc}}$ be $r$. Consider the partial order $<_{D_{\text{tc}}}$ implied by $D_{\text{tc}}$ on the vertex set $\mathcal{V}$. Define $\{T_i : i \in [r+1]\}$ as the unique partitioning of vertices of $D_{\text{tc}}$ where $T_i <_{D_{\text{tc}}} T_j, \forall i < j$ and each node in $T_i$ is a set of mutually incomparable elements. In other words, $T_i$ are the set of nodes at layer $i$ of the transitive closure graph $D_{\text{tc}}$. Define $\mathcal{T}_i = \cup_{k=1}^{i-1} T_k$. We have the following observation: $Pa_i \subset \mathcal{T}_i$. This paves the way for Algorithm 2 that leverages Lemma 3.

**Algorithm 2** LearnObservableGraph/Deterministic - Given the ancestral graph, access to a conditional independence testing oracle (CI oracle) and outputs the graph induced on observable nodes.

```
 1: function LEARNOBSERVABLEGRAPH/DETERIMINISTIC(M)
 2:     E = ∅.
 3:     for i in {r + 1, r, r − 1, . . . , 2} do
 4:         Intervene on the set Tᵢ of nodes.
 5:         Use samples from M_Tᵢ and use the CI-oracle to test the following.
 6:         for X in Tᵢ do
 7:             if (X ⊥̸ Y)_Dℓ[Tᵢ] then
 8:                 E ← E ∪ (X, Y).
 9:             end if
10:         end for
11:     end for
12:     return Observable graph
13: end function
```

The correctness of Algorithm 2 follows from Lemma 3, which is stated explicitly in the sequel.

**Theorem 2.** *Let $r$ be the length of the longest directed path in the causal graph $D_\ell$. Algorithm 2 requires only $r$ interventions and conditional independence tests on samples obtained from each one of the post-interventional distributions and outputs the observable graph $D$.*

## 4.2 A Randomized Algorithm

We propose a randomized algorithm that repeatedly uses the ancestor graph learning algorithm from Section 3 to learn the observable graph [5]. A key structure that we use is the transitive reduction:

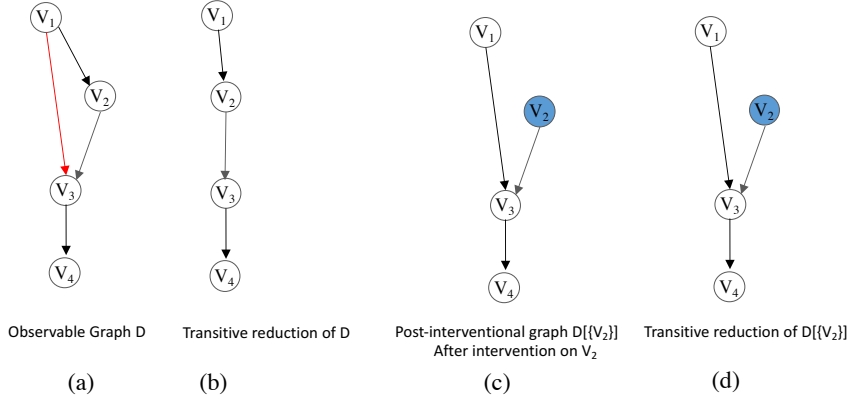

| Observable Graph D | Transitive reduction of D | Post-interventional graph D[{V₂}] After intervention on V₂ | Transitive reduction of D[{V₂}] |
| :---: | :---: | :---: | :---: |
| (a) | (b) | (c) | (d) |

Figure 1: Illustration of Lemma 5 - (a) An example of an observable graph $D$ without latents (b): Transitive reduction of $D$. The highlighted red edge $(V_1, V_3)$ has not been revealed under the operation of transitive reduction. c) Intervention on node $V_2$ and its post interventional graph $D[\{V_2\}]$ d) Since all parents of $V_3$ above $V_1$ in the partial order have been intervened on, by Lemma 5, the edge $(V_1, V_3)$ is revealed in the transitive reduction of $D[\{V_2\}]$.

**Definition 2** (Transitive Reduction). *Given a directed acyclic graph $D = (V, E)$, let its transitive closure be $D_{\mathrm{tc}}$. Then $\mathrm{Tr}(D) = (V, E_r)$ is a directed acyclic graph with minimum number of edges such that its transitive closure is identical to $D_{\mathrm{tc}}$.*

**Lemma 4.** *[1] $\mathrm{Tr}(D)$ is known to be unique if $D$ is acyclic. Further, the set of directed edges of $\mathrm{Tr}(D)$ is a subset of the directed edges of $D$, i.e., $E_r \subset E$. Computing $\mathrm{Tr}(D)$ from $D$ takes the same time as transitive closure of a DAG $D$, which takes time $\mathrm{poly}(n)$.*

We note that $\mathrm{Tr}(D) = \mathrm{Tr}(D_{\mathrm{tc}})$. Now, we provide an algorithm that outputs an observable graph based on samples from the post-interventional distribution after a sequence of interventions. Let us assume an ordering $\pi$ on the observable vertices $\mathcal{V}$ that satisfies the partial order relationships in the observable causal graph $D$. The key insight behind the algorithm is given by the following Lemma.

**Lemma 5.** *Consider an intervention on a set $S \subset \mathcal{V}$ of nodes in the observable causal graph $D$. Consider the post-interventional observable causal graph $D[S]$. Suppose for a specific observable node $V_i$, $V_i \in S^c$. Let $Y$ be a direct parent of $V_i$ in $D$ such that all the direct parents of $V_i$ above $Y$ in the partial order[6] $\pi(\cdot)$ is in $S$, i.e., $\{X : \pi(X) > \pi(Y), (X, V) \in D\} \subseteq S$. Then, $\mathrm{Tr}(D[S])$ will contain the directed edge $(Y, V_i)$ and it can be computed from $\mathrm{Tr}((D[S])_{\mathrm{tc}})$*

We illustrated Lemma 5 through an example in Figure 1. The red edge in Figure 1(a) is not revealed in the transitive reduction. The edge is revealed when computing the transitive reduction of the post-interventional graph $D[\{V_2\}]$. This is possible because all parents of $V_3$ above $V_1$ in the partial order (in this case node $V_2$) have been intervened on.

Lemma 5 motivates Algorithm 3. The basic idea is to intervene in randomly, then compute the transitive closure of the post-interventional graph using the algorithm in the previous section, compute the transitive reduction, and then accumulate all the edges found in the transitive reduction at every stage. We will show in Theorem 3 that with high probability, the observable graph can be recovered.

**Theorem 3.** *Let $d_{\max}$ be greater than the maximum in-degree in the observable graph $D$. Algorithm 3 requires at most $8cd_{\max}(\log n)^2$ interventions and CI tests on samples obtained from post-interventional distributions, and outputs the observable graph with probability at least $1 - \frac{1}{n^{c-2}}$.*

**Remark.** The above algorithm takes as input a parameter $d_{\max}$ that needs to be estimated. One practical option is to gradually increase $d_{\max}$ and run Algorithm 3.

**Algorithm 3** LearnObservable- Given access to a conditional independence testing oracle (CI oracle), a parameter $d_{\max}$ outputs induced subgraph between observable variables, i.e. $D$

---

1: **function** LEARNOBSERVABLE/RANDOMIZED($\mathcal{M}, d_{\max}$)
2:     $E = \emptyset$.
3:     **for** $i$ in $[1 : c * 4 * \mathrm{d}_{max} \log n]$ **do**
4:         $S = \emptyset$.
5:         **for** $V \in \mathcal{V}$ **do**
6:             $S \leftarrow S \cup V$ randomly with probability $1 - 1/d_{\max}$.
7:         **end for**
8:         $\hat{D}_S = \mathrm{LearnAncestralRelations}(\mathcal{M})$. Let $\hat{D} = (\mathcal{V}, \hat{E})$.
9:         Compute the transitive reduction of $\hat{D}(\mathrm{Tr}(\hat{D}_S))$ according to the algorithm in [1].
10:         Add the edges of the transitive reduction to the set $E$ if not already there, i.e. $E \leftarrow E \cup \hat{E}$.
11:     **end for**
12:     **return** The directed graph $(\mathcal{V}, E)$.
13: **end function**

---

## 5 Learning Latents from the Observable Graph

The final stage of our framework is learning the existence and location of latent variables given the observable graph. We divide this problem into two steps – first, we devise an algorithm that can learn the latent variables between any two variables that are non-adjacent in the observable graph; later, we design an algorithm that learns the latent variables between every pair of adjacent variables.

### 5.1 Baseline Algorithm for Detecting Latents between Non-edges

Consider two variables $X$ and $Y$ such that $X \leftarrow L \rightarrow Y$ and where $L$ is a latent variable. Clearly, to distinguish it from the case where $X$ and $Y$ are disconnected and have no latents, one needs check if $X \not\perp Y$ or not. This is a conditional independence test. For any non edge $(X, Y)$ in the observable graph $D$, when the observable graph $D$ is known, to check for latents between them, when other variables and possible confounders are around, one has to simply intervene on the rest of the $n - 2$ variables and do a independence test between $X$ and $Y$ in the post interventional graph. This requires a distinct intervention for every pair of variables. If the observable graph has maximum degree $d = o(n)$, this requires $\Theta(n^2)$ interventions. We will reduce this to $O(d^2 \log n)$ interventions which is an exponential improvement for constant degree graphs.

### 5.2 Latents between Non-adjacent Nodes

We start by noting the following fact about causal systems with latent variables:

**Theorem 4.** *Consider two non-adjacent nodes $X_i, X_j$. Let $S$ be the union of the parents of $X_i, X_j$, $S = Pa_i \cup Pa_j$. Consider an intervention on $S$. Then we have $(X_i \not\perp X_j)_{\mathcal{M}_S}$ if and only if there exists a latent variable $L_{i,j}$ such that $X_j \leftarrow L_{i,j} \rightarrow X_i$. The statement holds under an intervention $S$ such that $Pa_i \cup Pa_j \subset S$, $X_i, X_j \notin S$.*

The above theorem motivates the following approach: For a set of nodes which forms an independent set, an intervention on the union of parents of the nodes of the independent set allows us to learn the latents between any two nodes in the independent set. We leverage this observation using the following lemma on the number of such independent sets needed to cover all non-edges.

**Lemma 6.** *Consider a directed acyclic graph $D = (V, E)$ with degree (out-degree+in-degree) $d$. Then there exists a randomized algorithm that returns a family of $m = \mathcal{O}(4e^2(d + 1)^2 \log(n))$ independent sets $\mathcal{I} = \{I_1, I_2, \ldots, I_m\}$ that cover all non-edges of $D$: $\forall i, j$ such that $(X_i, X_j) \notin E$ and $(X_j, X_i) \notin E$, $\exists k \in [m]$ such that $X_i \in I_k$ and $X_j \in I_k$, with probability at least $1 - \frac{1}{n^2}$.*

Note that this is a randomized construction and we are not aware of any deterministic construction. Our deterministic causal learning algorithm requires oracle access to such a famiy of independent sets, whereas our randomized algorithm can directly use this randomized construction. Now, we use this observation to construct a procedure to identify latents between non-edges (see Algorithm 4). The following theorem about its performance follows from Lemma 6 and Theorem 4.

**Algorithm 4** LearnLatentNonEdge- Given access to a CI oracle, observable graph $D$ with max degree $d$ (in-degree+out-degree), outputs all latents between non-edges

```
 1: function LEARNLATENTNONEDGE(M, d_max)
 2:     L = ∅.
 3:     Apply the randomized algorithm in Lemma 6 to find a family of independent sets I =
        {I_1, I_2, ..., I_m} that cover all non-edges in D such that m ≤ 4e²(d + 1)² log(n).
 4:     for j ∈ [1 : m] do
 5:         Intervene on the parent set of the nodes in I_j.
 6:         for every pair of nodes X, Y in I_j  do
 7:             if (X ⊥̸ Y)_{D_ℓ[I_j]} then
 8:                 L ← L ∪ {X, Y}.
 9:             end if
10:         end for
11:     end for
12:     return The set of non-edges L.
13: end function
```

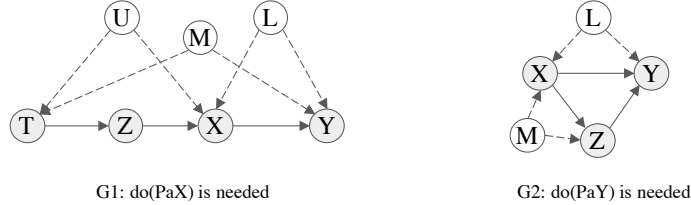

G1: do(PaX) is needed          G2: do(PaY) is needed

Figure 2: Left: A graph where intervention on the parents of $X$ is needed for do-see test to succeed. Right: A graph where intervention on the parents of $Y$ is needed for do-see test to succeed.

**Theorem 5.** *Algorithm 4 outputs a list of non-edges $L$ that have latent variables between them, given the observable graph $D$, with probability at least $1 - \frac{1}{n^2}$. The algorithm requires $4e^2(d+1)^2 \log(n)$ interventions where $d$ is the max-degree (in-degree+out-degree) of the observable graph.*

### 5.3 Latents between Adjacent Nodes

We construct an algorithm that can learn latent variables between the variables adjacent in the observable graph. Note that the approach of CIT testing in the post-interventional graph is not helpful. Consider the variables $X \to Y$. To see the effect of the latent path, one needs to cut the direct edge from $X$ to $Y$. This requires intervening on $Y$. However, such an intervention disconnects $Y$ from its latent parent. Thus we resort to a different approach compared to the previous stages and exploit a different characterization of causal Bayesian networks called a 'do-see' test.

A do-see test can be described as follows: Consider again a graph where $X \to Y$. If there are no latents, we have $\mathbb{P}(Y|X) = \mathbb{P}(Y|\text{do}(X))$. Assume that there is a latent variable $Z$ which causes both $X$ and $Y$, then excepting the pathological cases[7], $\mathbb{P}(Y|X) \neq \mathbb{P}(Y|\text{do}(X))$.

Figure 2 illustrates the challenges associated with a do-see test in bigger graphs with latents. Graphs $G1$ and $G2$ are examples where parents of both nodes involved in the test need to be included in the intervention set for the Do-see test to work. In $G1$, suppose we condition on $X$, as required by the 'see' test. This opens up a non-blocking path $X - U - T - M - Y$. Since $X \to Y$ is not the only d-connecting path, it is not necessarily true that $\mathbb{P}(Y|X) = \mathbb{P}(Y|\text{do}(X))$. Now suppose we perform the do-see test under the intervention $\text{do}(Z)$. Then the aforementioned path is closed since $X$ is not a descendant of $T$ in the post interventional graph. Hence we have $\mathbb{P}(Y|X, \text{do}(Z)) = \mathbb{P}(Y|\text{do}(X, Z))$. Similarly $G2$ shows that intervening on the parent set of $Y$ is also necessary.

We have the following theorem, which shows that we can perform the do-see test between $X, Y$ under $\text{do}(Pa_X, Pa_Y)$:

**Theorem 6.** *[Interventional Do-see test] Consider a causal graph $D$ on the set of observable variables $\mathcal{V} = \{V_i\}_{i \in [n]}$ and latent variables $L = \{L_i\}_{i \in [m]}$ with edge set $E$. If $(V_i, V_j) \in E$, then*

$$\Pr(V_j | V_i = v_i, do(Pa_i = pa_i, Pa_j = pa_j)) = \Pr(V_j | do(V_i = v_i, Pa_i = pa_i, Pa_j = pa_j)),$$

*iff $\nexists k$ such that $(L_k, V_i) \in E$ and $(L_k, V_j) \in E$, where $Pa_i$ is the set of parents of $V_i$ in $V$. Quantities on both sides are invariant irrespective of additional interventions elsewhere.*

Next we need a subgraph structure to perform multiple do-see tests at once in order to efficiently discover the latents between the adjacent nodes. Performing the test for every edge would take $\mathcal{O}(n)$ even in graphs with constant degree. We use strong edge coloring of sparse graphs.

**Definition 3.** *A strong edge coloring of an undirected graph with $k$ colors is a map $\chi : E \to [k]$ such that every color class is an induced matching. Equivalently, it is an edge coloring such that any two nodes adjacent to distinct edges with the same color are non-adjacent.*

Graphs of maximum degree $d$ can be strongly edge-colored with at most $2d^2$ colors.

**Lemma 7.** *[6] A graph of maximum degree $d$ can be strongly edge-colored with at most $2d^2$ colors. A simple greedy algorithm that colors edges in sequence achieves this.*

Now observe that a color class of the edges forms an induced matching. We show that due to this, the 'do' part (RHS of Theorem 6) of all the do-see tests in a color class can be performed with a single intervention while the 'see' part (RHS of Theorem 6) can be again performed with another intervention. We argue that we need exactly two different interventions per color class. The following theorem uses this property to prove correctness of Algorithm 5.

---

**Algorithm 5** LearnLatentEdge- Observable graph $D$ with max degree $d$ (in-degree+out-degree), outputs all latents between edges

---

1: **function** LEARNLATENTEDGE($\mathcal{M}, d$)
2:     $L = \emptyset$.
3:     Apply the greedy algorithm in Lemma 7 to color the edges of $D$ with $k \leq 2d^2$ colors.
4:     **for** $j \in [1 : k]$ **do**
5:         Let $A_j$ be the nodes involved with the edges that form color class $j$. Let $P_j$ be the union of parents of all nodes in $A_j$ except the nodes in $A_j$.
6:         Let the set of tail nodes of all edges be $T_j$.
7:         Following loop requires the intervention on the set $T_j \cup P_j$, i.e. $do(\{T_j, P_j\})$.
8:         **for** Every directed edge $(V_t, V_h)$ in color class $j$ **do**
9:             Calculate $S(V_t, V_h) = P(V_h | do(T_j, P_j))$ using post interventional samples.
10:         **end for**
11:         Following loop requires the intervention on the set $P_j$.
12:         **for** Every directed edge $(V_t, V_h)$ in color class $j$ **do**
13:             Calculate $S'(V_t, V_h) = P(V_h | V_t, do(P_j))$ using post interventional samples.
14:             **if** $S'(V_t, V_h) \neq S(V_t, V_h)$ **then**
15:                 $L \leftarrow L \cup (V_t, V_h)$
16:             **end if**
17:         **end for**
18:     **end for**
19:     **return** The set of edges $L$ that have latents between them.
20: **end function**

---

**Theorem 7.** *Algorithm 5 requires at most $4d^2$ interventions and outputs all latents between the edges in the observable graph.*

## 6 Conclusions

Learning cause-and-effect relations is one of the fundamental challenges in science. We studied the problem of learning causal models with latent variables using experimental data. Specifically, we introduced two efficient algorithms capable of learning direct causal relations (instead of ancestral relations) and finding the existence and location of potential latent variables.

## Footnotes

[2]Hereafter, *latent variable* refers to any unmeasured variable that affects more than one observed variable.

[3]We assume access to an oracle that outputs a size-$\mathcal{O}(d^2 \log(n))$ independent set cover for the non-edges of a given graph. This oracle can be implemented using another randomized algorithm as we explain in Section 5.

[4]For convenience, detailed definitions of blocking and non-blocking paths are provided in the full version [20].

[5]Note that this algorithm does not require learning the ancestral graph first.

[6]The nodes above with respect to the partial order of a graph are those that are closer to the source nodes.

[7]These cases are fully identified in the full version [20].

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
