[Reviews · NeurIPS 2017]

Reviewer 1



The authors propose theory and algorithms for identifying ancestral relations, causal edges and latent confounders using hard interventions. Their algorithms assume that it is possible to perform multiple interventions on any set of variables of interest, and the existence of an independence oracle, and thus is mostly of theoretical value. In contrast to previous methods, the proposed algorithms do not assume causal sufficiency, and thus can handle confounded systems. The writing quality of the paper is good, but some parts of it could be changed to improve clarity. General Comments Several parts of the paper are hard to follow (see below). Addressing the below comments should improve clarity. In addition, it would be very helpful if further examples are included, especially for Algorithms 1-3. Furthermore, results are stated but no proof is given. Due to lack of space, this may be challenging. For the latter, it would suffice to point to the appendix provided in the supplementary material. Some suggestions to save space: (1) remove the end if/for/function parts of the algorithms, as using appropriate indentation suffices, (2) the “results and outline of the paper” section could be moved to the end of the introduction, and several parts in the introduction and background could be removed or reduced. Although existing work is mentioned, the connections to previous methods are not always clear. For instance, how do the proposed methods relate to existing methods that assume causal sufficiency? How about methods that do not assume causal sufficiency, but instead make other assumptions? Also, the paper [1] (and references therein) are related to the work but not mentioned. In the abstract, the authors mention that some experiments may not be technically feasible or unethical, yet this is not addressed by the proposed algorithms. In contrast, the algorithms assume that any set of interventions is possible. This should be mentioned as a limitation of the algorithms. Furthermore, some discussion regarding this should be included. An important question is if and how such restrictions (i.e. some variables can’t be intervened upon) affect the soundness/completeness of the algorithms. Lines 47-50: The statement regarding identifiability of ancestral relations is incorrect. For some edges in MAGs it can be shown that they are direct, or whether the direct relation is also confounded; see visible edges in [2] and direct causal edges in [3]. If shown to be direct, they are so only in the context of the measured variables; naturally, multiple variables may mediate the direct relation. Line 58: [4] is able to handle interventional data with latent variables. Other algorithms that are able to handle experimental data with latent variables also exist [5-9]. Definition 1: How is the strong separating system computed? (point to appendix from the supplementary material) Lemma 3: Shouldn’t also X_i \not\in S? Algorithm 2, line 7: What is Y? This point is crucial for a correct understanding of the algorithm. I assume there should be an iteration over all Y that are potential children of X, but are not indirect descendants of X. Lemma 4: For sparse graphs, the transitive closure can be trivially computed in O(n * (n + m)) time (n: number of nodes, m: number of edges) which is faster than O(n^\omega). I mention this as the graphs considered in this paper are sparse (of constant degree). Line 179: The fact that Tr(D) = Tr(D_{tc}) is key to understanding Lemma 5 and Algorithm 3 and should be further emphasized. Algorithm 3: It is not clear from the algorithm how S is used, especially in conjunction with Algorithm 1. One possibility is for Algorithm 1 to also always intervene on S whenever intervening on set S_i (that is, intervene on S_i \cup S). Is this correct? This part should be improved. Algorithm 3: Using S to intervene on a random variable set, and then performing additional interventions using Algorithm 1 raises the question whether a more efficient approach exists which can take into consideration the fact that S has been intervened upon. Any comments on this? Algorithm 3: It would be clearer if the purpose of c is mentioned (i.e. that it is a parameter which controls the worst case probability of recovering the observable graph). Line 258: Induced matching is not defined. Although this is known in graph theory, it should be mentioned in the paper as it may confuse the reader. Do-see test: Although not necessary, some comment regarding the practical application of the do-see test would be helpful, if space permits. Typos / Corrections / Minor Suggestions Pa_i is used multiple times but not defined The longest directed path is sometimes denoted as r and sometimes as l. Use only one of them. Abstract: O(log^2(n)) -> O(d*log^2(n)) Line 30: … recovering these relations … -> … recovering some relations … (or something along that lines, as they do not identify all causal relations) Line 59: international -> interventional Line 105: Did you mean directed acyclic graph? Line 114: (a) -> (b) Line 114: … computed only transitive closures … -> … computing using transitive closured … Line 120: (a) -> (c) Line 159: Define T_i … -> something wrong here Line 161: Using a different font type for T_i is confusing. I would recommend using different notation. Lemma 5, Line 185: V_i \in S^c would be clearer if written as V_i \not in S (unless this is not the same and I missed something) Line 191: We will show in Lemma 5 -> Shouldn’t it be Theorem 3? Algorithm 3: \hat{D}(Tr(\hat{D}_S)) looks weird. Consider removing Tr(\hat{D}_S) (unless I misunderstood something). Lines 226-227: To see the effect of latent path Line 233: Figure 4.2 -> Figure 2 Line 240: … which shows that when … -> remove that [1] Hyttinen et al, Experiment Selection for Causal Discovery, JMLR 2013 [2] Zhang, Causal Reasoning with Ancestral Graphs, JMLR 2008 [3] Borboudakis et al, Tools and Algorithms for Causally Interpreting Directed Edges in Maximal Ancestral Graphs, PGM 2012 [4] Triantafillou and Tsamardinos. Constraint-based causal discovery from multiple interventions over overlapping variable sets. JMLR 2015 [5] Hyttinen et al, Causal Discovery of Linear Cyclic Models from Multiple Experimental Data Sets with Overlapping Variables, UAI 2012 [6] Hyttinen et al, Discovering Cyclic Causal Models with Latent Variables: A General SAT-Based Procedure, UAI 2013 [7] Borboudakis and Tsamardinos, Towards Robust and Versatile Causal Discovery for Business Applications, KDD 2016 [8] Magliacane et al, Ancestral Causal Inference, NIPS 2016 [9] Magliacane et al, Joint Causal Inference from Observational and Experimental Datasets, Arxiv